# Exploring Alternative Treatment Choices for Multidrug-Resistant Clinical Strains of *Helicobacter pylori* in Mongolia

**DOI:** 10.3390/microorganisms11122852

**Published:** 2023-11-24

**Authors:** Ayush Khangai, Batsaikhan Saruuljavkhlan, Dashdorj Azzaya, Boldbaatar Gantuya, Khasag Oyuntsetseg, Junko Akada, Takashi Matsumoto, Yoshio Yamaoka

**Affiliations:** 1Department of Environmental and Preventive Medicine, Oita University Faculty of Medicine, Yufu 879-5593, Japan; m20d9103@oita-u.ac.jp (A.K.); saruuljavkhlan@yahoo.com (B.S.); akadajk@oita-u.ac.jp (J.A.); tmatsumoto9@oita-u.ac.jp (T.M.); 2The Gastroenterology Center, The First Central Hospital of Mongolia, Ulaanbaatar 14210, Mongolia; 3Endoscopy Unit, Department of Gastroenterology, Mongolia Japan Hospital of Mongolian National University of Medical Sciences, Ulaanbaatar 14210, Mongolia; medication_bg@yahoo.com (B.G.); oyuntsetseg.kh@mnums.edu.mn (K.O.); 4Department of Gastroenterology, Mongolian National University of Medical Sciences, Ulaanbaatar 14210, Mongolia; azzaya2000@gmail.com; 5Division of Gastroentero-Hepatology, Department of Internal Medicine, Faculty of Medicine-Dr. Soetomo Teaching Hospital, Universitas Airlangga, Surabaya 60115, Indonesia; 6Research Center for GLOBAL and LOCAL Infectious Diseases, Oita University, Yufu 879-5593, Japan; 7Department of Medicine-Gastroenterology, Baylor College of Medicine, Houston, TX 77030, USA

**Keywords:** *Helicobacter pylori*, antibiotic resistance, ciprofloxacin, moxifloxacin, rifabutin, furazolidone, next-generation sequencing, resistant gene mutation, Mongolia

## Abstract

*Helicobacter pylori* is a pathogen related to severe diseases such as gastric cancer; because of rising antimicrobial-resistant strains, failure to eradicate *H. pylori* with antibiotics has increased worldwide. Multidrug-resistant *H. pylori* and gastric cancer is common in Mongolia; therefore, we aimed to explore alternative antimicrobial treatments and the genomes of resistant strains in this country. A total of 361 *H. pylori* strains isolated from patients in Mongolia were considered. Minimal inhibitory concentrations for two fluoroquinolones (ciprofloxacin and moxifloxacin), rifabutin, and furazolidone were determined via two-fold agar dilution. Genomic mutations in antibiotic-resistant strains were identified by next-generation sequencing using the Illumina Miseq platform and compared with genes from a reference *H. pylori* strain (26695). The resistance rate of *H. pylori* strains to quinolones was high (44% to ciprofloxacin and 42% to moxifloxacin), and resistance to rifabutin was low (0.5%); none were resistant to furazolidone. Most quinolone-resistant strains possessed *gyrA* gene mutations causing amino acid changes (e.g., N87K, A88P, and D91G/Y/N). While one rifabutin-resistant strain had amino acid-substituting mutations in *rpoB* (D530N and R701C), the other had three novel *rpoB* mutations; both rifabutin-resistant strains were sensitive to furazolidone. Overall, our findings suggest that rifabutin and/or furazolidone may be an alternative, effective *H. pylori* treatment in patients who have failed to respond to other treatment regimens.

## 1. Introduction

Over the last 40 years, antibiotic resistance to *Helicobacter pylori* has increased globally, with higher rates in less-developed countries [1]. Because of its resistance rate, *H. pylori* has emerged as one of sixteen antibiotic-resistant pathogens that pose the most serious threat to human health, according to the World Health Organization [2,3,4]. Both the American College of Gastroenterology and the European Helicobacter and Microbiota Study group have changed their guidelines regarding *H. pylori* infection to offer the treatment for positive cases; in addition, *H. pylori* was included in the latest version of the World Health Organization’s International Classification of Diseases (ICD-11) as a cause of gastritis [5].

The most popular treatment regimens for *H. pylori* use a combination of two antibiotics (amoxicillin, clarithromycin, metronidazole, or tetracycline) with a proton-pump inhibitor (PPI) and/or bismuth subcitrate [6]. According to the Maastricht VI/Florence consensus report, first-line treatment efficacy is more than 90% for individuals who are being treated for the first time. However, if an individual fails to respond to the first-line treatment regime, secondary antibiotics or alternative antibiotics may be required. Currently, rifabutin- and/or furazolidone-based regimens (e.g., ciprofloxacin, moxifloxacin, levofloxacin) are used when eradication fails multiple times because of fluoroquinolone resistance [7].

Antimicrobial resistance mechanisms to fluoroquinolones, nitrofurans (e.g., furazolidone), and rifamycin derivatives (e.g., rifabutin) are well studied, especially in *H. pylori*. Fluoroquinolones exert their bactericidal activity by inhibiting two essential bacterial type II topoisomerases, DNA gyrase (Gyr) and topoisomerase IV; both are heterotetrametric enzymes consisting of two A subunits and two B subunits, and modulate the chromosomal supercoiling required for DNA synthesis, transcription, and cell division [8,9]. Target-mediated resistance due to mutations on the GyrA or/and GyrB subunits of gyrase is one of the main reasons for fluoroquinolone regimen failure [10], with resistance occurring because of either a single or dual mutations in the *gyrA* or *gyrB* codon regions [11,12]. We, and others, previously determined that amino acid-substituting mutations in the *gyrA* gene (N87K, D91N, D91G, D91Y) were frequently observed in highly resistant *H. pylori* strains [13,14]; however, there are some resistant strains without identified mutations. Apart from fluoroquinolones, furazolidone is increasingly used to treat *H. pylori* infection. Furazolidone primarily works through 2-oxoglutarate oxidoreductase subunit D and/or pyruvate–ferredoxin oxidoreductase subunit D to bind free radicals and counteract DNA replication and protein production [15,16]. Finally, rifabutin inhibits the DNA-dependent RNA polymerase of *H. pylori* at very low concentrations in vitro [17,18]; resistance to rifabutin is related to amino acid substitution in the β subunit of RNA polymerase (RpoB).

In Mongolia, in recent studies, it was reported that multidrug resistance (MDR) in *H. pylori* strains is high: 51% for amoxicillin, clarithromycin, metronidazole, levofloxacin, and minocycline [14] and 60.8% for amoxicillin, clarithromycin, metronidazole, erythromycin, and nitrofuran [13]. Because of the high rate of MDR, alternative drugs to treat *H. pylori* are needed. In this study, we aimed to investigate the efficacy of new antimicrobials against *H. pylori* strains with MDR; specifically, we studied ciprofloxacin (CIP), moxifloxacin (MOXI), furazolidone (FUR), and rifabutin (RFB). We also aimed to define genomic mutations underlying antibiotic resistance using next-generation sequencing analysis of *H. pylori*. Overall, we found that furazolidone and rifabutin may be key to treating *H. pylori* strains with MDR in Mongolia.

## 2. Results

### 2.1. Sample and Antibiotic Resistance Distribution

All 361 *H. pylori* strains previously reported by our group [14] were identified in this study. The average age of the participants was 44.3 ± 13.4 (mean ± SD) years, and 26.9% (97/361) were male. The resistance status of all strains to seven antibiotics is shown in Table 1.

Initially, we selected 91 representative *H. pylori* strains from a total of 361 strains (25% of all strains), which were selected based on five antimicrobial resistance statuses identified previously [14]: 20 strains sensitive to all antibiotics, 21 strains resistant to a single antibiotic, 20 strains resistant to two antibiotics, 20 strains resistant to three antibiotics, and 10 strains resistant to five antibiotics. These 91 strains were used to determine MICs for the quinolone group antibiotics CIP and MOXI, which were compared with a previously reported MIC for levofloxacin (LVF). We found that *H. pylori* strains exhibited a similar level of resistance to all three quinolone group antibiotics (LVF, 44%; CIP, 43%; and MOXI, 42%), which was consistent across sexes (Table 2 and Figure 1).

Among the 91 tested strains, 49 were sensitive to quinolones: 42 to all three and 7 to one or two (LVF-R, MOXI-S, CIP-S: two strains; LVF-R, MOXI-R, CIP-S: one strain; LVF-R, MOXI-S, CIP-R: two strains; LVF-S, MOXI-R, CIP-R: two strains).

Two further antibiotics, FUR and RFB, were tested in these 91 strains. We found no resistance to FUR; however, one strain was resistant to RFB (strain UB217, MIC = 4.0 µg/mL). To find whether there was more resistance to RFB or not and to FUR against *H. pylori* strains, MICs for FUR and RFB were then determined for the remaining 270 strains. Tests for all 361 strains showed that resistance rates to FUR and RFB were very low: 0% (no strains) and 0.5% (2 strains), respectively. MICs for FUR were determined as <0.064 µg/mL (1 strain), 0.125 µg/mL (15 strains), 0.25 µg/mL (236 strains), and 0.5 µg/mL (109 strains). In all 361 *H. pylori* strains, FUR had an MIC less than the 4 µg/mL cut-off required to define antimicrobial resistance (Figure 2).

For all 361 *H. pylori* strains, RFB had a MIC of 0.03 µg/mL against 136 strains (37.6%) and an MIC of 0.008 µg/mL against 122 strains (33.7%). Two strains showed RFB resistance (UB217, MIC 4.0 µg/mL and Ke9, MIC 8 µg/mL) (Figure 3).

Next, we focused on whether these two RFB-resistant strains were multiple-antibiotic-resistant. Strain Ke9 was resistant to four other antibiotics, MNZ and the three quinolones, but was sensitive to AMX, CLA, MINO, and FUR. Strain UB217 was resistant to six other antibiotics, but sensitive to MINO and FUR (Table 3).

### 2.2. Genotypic Determination of Antibiotic Resistance

Separately from the agar dilution tests in this study, we performed full-genome sequencing of 229 *H. pylori* strains (63%), including 77 (85%) sequences from the representative 91 strains. Protein sequence alignment was performed using these 77 strains for genes related to quinolone resistance and all 229 strains for genes related to rifabutin resistance. A total of 10 genomes from the 10 strains sensitive to all nine antibiotics and a partial genome from reference strain *H. pylori* 26695 were used as reference gene sequences. The potential genotypes of quinolone- and rifamycin-resistant *H. pylori* strains are shown in Appendix A.

### 2.3. Quinolone Resistance

Within the *gyrA* gene of the 77 sequences assessed for quinolone resistance, previously reported genetic variations, such as N87K/I and D91G/Y/N substitutions, were frequently observed, although the A88P substitution was found in only one strain (Table 4). The amino acid substitutions N87K/I (*p* < 0.0001) and D91N/Y/G (*p* < 0.0001) were significantly associated with LVF, MOXI, and CIP resistance, according to Fisher’s exact test (Appendix A). We also found N87K and D91G mutations in some strains that had mixed quinolone-resistant status. Of the 39 quinolone-resistant strains, gyrase subunit gene analysis showed that 39 and 30 hypothetical mutations related to quinolone-resistant phenotypes existed in *gyrA* and *gyrB*, respectively (Appendix A). Of these 39 hypothetical *gyrA* mutations, V199I (*p* < 0.012) and N660D (*p* < 0.005) were significantly associated with resistance in the strains Ke86, Kh95, Ub72, Uvs113, and Ub72. However, for eight resistant strain sequences, there were no significant mutations in the quinolone resistance-determining region of *gyrB*_E415-S454_ and *gyrA*_A71-Q110_.

### 2.4. Rifabutin Resistance

Within the *rpoB* gene of the 229 sequences assessed for rifabutin resistance, we found the previously reported mutations D530N and R701C in strain UB217 (MIC 4.0 µg/mL), while strain Ke9 had multiple hypothetical mutations. In addition, we found the previously reported V538I substitution in two strains (UB75 and Kh67). However, these strains were sensitive to RFB with MICs of 0.12 and 0.06 µg/mL, respectively. There were multiple putative mutations noted, such as S273P, E470G, A1173T, L2196P, and A2710V, which are shown in Table 5 and Appendix A (green letters).

## 3. Discussion

In this study, we aimed to find new drugs to treat MDR *H. pylori* strains. To do this, we selected 91 representative resistant strains based on resistance status from a Mongolian population and found very high resistance rates to CIP (43%) and MOXI (42%), similar to rates for LVF (44%) previously reported. A recent retrospective study from China, as a neighboring country, revealed that the quinolone resistance rate is similar: around 48% within over 3300 clinical isolates [19]. Next-generation sequencing data from 39 quinolone-resistant and 38 quinolone-sensitive *H. pylori* strains showed that D91 and N87 *gyrA* gene mutations commonly existed in the resistant strains, in line with previous studies [4,10,14,19]. We also identified two mutations (V119I and N660D) that were significantly associated with resistance in strains without any well-known mutations. In contrast, we found that *H. pylori* strains were highly sensitive to both FUR and RFB, with sensitivity rates of 100% and 99.5%, respectively, even in MDR strains. Therefore, while the three quinolone antibiotics do not appear to be viable alternative antibiotic treatments in Mongolia, FUR and RFB are promising candidates.

Despite their efficacy in the present study, RFB- and/or FUR-based antibiotic treatments are only appropriate for individuals who do not respond to more than one standard treatment course, rather than as a first-line treatment. According to the literature, multiple codons of the *rpoB* gene contain mutations resulting in strong resistance to antibiotics; these usually lie in the codon positions 149, 524–545, 585–586, and 701 [14,18,20]. Although we found some mutations in these locations, such as D540N and R701C in strain UB217, there were no such mutations in strain Ke9; however, three hypothetical mutations (E470G, L2196P, and A2710V) were identified. Moreover, we found a previously described mutation (D538I) in two RFB-sensitive strains (UB75 and Kh67), with MICs of 0.12 µg/mL and 0.06 µg/mL, respectively. The two RFB-resistant strains, Ke9 and UB217, were resistant to five and seven antibiotics, respectively; however, both strains were sensitive to FUR and MINO (Table 3). This suggests that there are still antibiotics able to effectively treat MDR *H. pylori* strains; however, appropriate use of these drugs is required to prevent the development of *H. pylori* strains resistant to them, as FUR-resistant strains have begun to be reported [21].

According to the Maastricht VI/Florence consensus report, developing countries tend to use quadruple therapy for *H. pylori* more frequently than developed countries because of higher rates of CLR resistance. Mongolia has high CLR resistance (29.9%), with *H. pylori* eradication rates of 68.5–97.6% [22], meaning eradication failure varies between 2.4% and 31.5%, depending on the treatment regimen [22]. In cases of *H. pylori* eradication failure, an RFB triple regimen (150 mg or 300 mg bis in die (BID, i.e., twice a day) or quarter in die (QID, i.e., four times a day)) is usable with AMX (1000 mg, BID) and a PPI (standard dose, BID) for 10–14 days. For FUR, triple or quadruple regimens (100–200 mg, BID) are useable with AMX (1000 mg, BID) or tetracycline (750 mg, BID) and a PPI (standard dose, BID) with or without bismuth subcitrate (220 mg, BID) for 7–14 days [7,10]. Therefore, we should think about the limitations of the above antibiotics. For instance, RFB is quite expensive, and sometime causes myelotoxicity, myalgia, rashes, etc., although these symptoms are rare [23].However, it is important to note that, in accordance with previous studies, RFB- and FUR-containing therapies should be recommended for individuals who have not responded to multiple (two or more) previous therapies [23,24,25,26,27].

## 4. Conclusions

Overall, our results suggest that in patients who do not respond to first- and/or second-line antibiotic regimens for *H. pylori* infection, an alternative treatment regimen with rifabutin and/or furazolidone may successfully treat infection and reduce the risk of further complications, such as gastric cancer. These drugs have no cross-resistance to metronidazole and are effective in populations with high metronidazole resistance. Our results also suggest that if patients do not respond to a particular antibiotic, further treatment with antibiotics from the same group (e.g., quinolones) should not be attempted because resistance is likely. Although finding drugs to eradicate MDR *H. pylori* is important, it is key that doctors ensure patients take rifabutin and furazolidone appropriately in order not to allow the development of strains resistant to these antibiotics. To ensure the widespread eradication of *H. pylori*, further studies on the adverse effects of rifabutin and furazolidone, and the potential of resistance to these drugs in susceptible countries, are required.

## 5. Materials and Methods

### 5.1. Study Population

In total, 1004 volunteers from Mongolia with dyspeptic symptoms participated in the study between November 2014 and August 2016. This sample was drawn from Ulaanbaatar (Mongolia’s capital city, in the central region), Khuvsgul province (in the north), Umnugovi province (in the south), Uvs province (in the West), and Khentii province (in the east). An upper gastrointestinal endoscopy was performed, and a biopsy sample was collected from the antrum for *H. pylori* culture; the specimen was placed in the transfer medium and then immediately frozen. Participants with a history of partial or total gastrectomy, or who had taken H_2_-receptor blockers or PPIs within four weeks of the endoscopy, were excluded from the study. Samples for culture were transferred to Ulaanbaatar and immediately stored at −80 °C. Later, all samples were transferred for *H. pylori* culture and further analysis at the Department of Environmental and Preventive Medicine, Oita University Faculty of Medicine, Japan.

### 5.2. Isolation and Culture

From biopsy samples, *H. pylori* samples were cultured on *H. pylori*-selective plates (Nissui Pharmaceutical Co., Ltd., Tokyo, Japan) and incubated at 37 °C under microaerophilic conditions (10% O_2_, 5% CO_2_, and 85% N_2_) for 3–7 days. Small, round, purple colonies typical of *H. pylori* were checked for Gram staining and tested for oxidase, catalase, and urease, as previously described [14].

### 5.3. Antibiotic Susceptibility Test

At first, we selected 91(25%) strains randomly from a total of 360 previous MIC determined strains. All tests were performed using the serial two-fold agar dilution method with MOXI, CIP, FUR, and RFB to determine the minimal inhibitory concentration (MIC), as previously described [14]. Briefly, bacteria were sub-cultured in Mueller Hinton II Agar medium (Becton Dickinson, New York, NY, USA) supplemented with 5% defibrinated horse blood. The bacterial suspension was adjusted to OD 0.1 using a OD_590_ spectrophotometer (UV-1800, Shimadzu, Kyoto, Japan), and a 48-pin inoculator was used to inoculate the culture plate with twofold serial dilutions of the test antibiotics. After 3–7 days of incubation (*H. pylori* microaerophilic conditions), the MIC of each antibiotic was determined as the minimal dilution at which bacterial colonies were no longer produced, according to the Clinical & Laboratory Standards Institute’s 11th edition of “Methods for Dilution Antimicrobial Susceptibility Tests for Bacteria That Grow Aerobically” (M07). *H. pylori* 26695 and DMSO were used as positive and negative controls and MICs were tested twice for each strain. Amoxicillin was used to compare the current results with those previously determined for these *H. pylori* strains; there was 97.5% agreement between results [14]. The reported clinical breakpoints for the antibiotics used in this study are 1.0 µg/mL for CIP and MOXI [27], 4.0 µg/mL for FUR [27,28,29], and 1.0 µg/mL for RFB [27]; each of these breakpoints was also used in this study.

### 5.4. DNA Extraction and Next-Generation Sequencing

A commercially available DNA purification kit (Qiagen, Hilden, Germany) was used to extract DNA from *H. pylori* on day 3 of culture. DNA was quantified using a Quantus Fluorometer (Promega, Madison, WI, USA), and sequenced using the MiSeq platform (Illumina, Inc., San Diego, CA, USA). Sequences were aligned with *gyrA* and *gyrB* for samples treated with CIP and MOXI and *rpoB* for samples treated with RFB based on standard *H. pylori* 26695 sequences (GenBank accession number NC_000915.1) using the CLC Genomics Workbench (v22.0, QIAGEN). Finally, genetic mutations in antimicrobial-related genes in each *H. pylori* strain were identified from the DNA sequences obtained under MiSeq sequencing. 

### 5.5. Statistical Analysis

SPSS Statistics version 25.0 (SPSS Inc., Chicago, IL, USA) was used for statistical analysis in our study. The chi-square test and Fisher’s exact test were used; *p* < 0.05 indicated a significant difference.

### 5.6. Nucleotide Sequence Accession Numbers

All nucleotide sequences analyzed in this study were deposited in the DDBJ Center Data Bank of Japan; accession numbers for the *gyrA* and *gyrB* genes are LC782993–LC789146, and for the *rpoB* gene LC789147–LC783375, respectively.

## Figures and Tables

**Figure 1 microorganisms-11-02852-f001:**
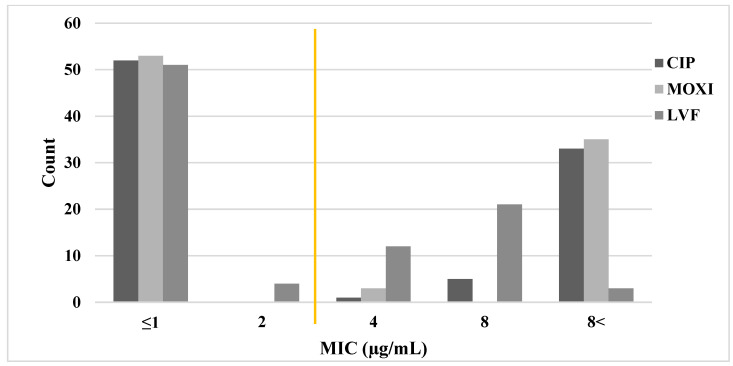
Minimal inhibitory concentration (MIC) distributions for moxifloxacin (MOXI), ciprofloxacin (CIP), and levofloxacin (LVF) against *H. pylori*. MIC (µg/mL) was determined by agar dilution assays of 91 representative Mongolian *H. pylori* strains. The MIC for levofloxacin was determined previously [14]. The yellow line represents the cutoff for resistance (≤2 µg/mL, sensitive; >2 µg/mL, resistant).

**Figure 2 microorganisms-11-02852-f002:**
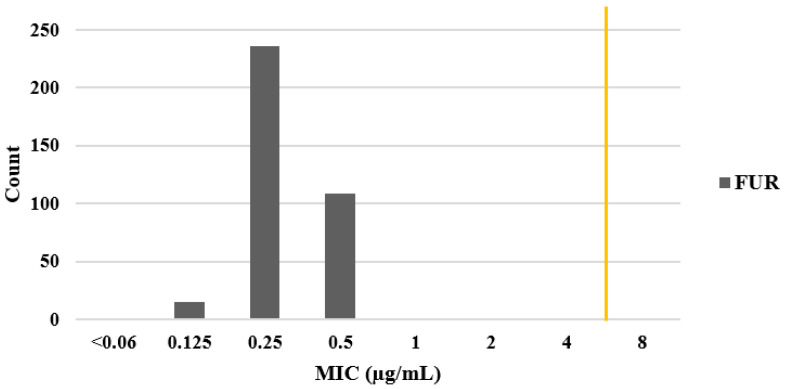
Minimal inhibitory concentration (MIC) distribution for furazolidone (FUR). MIC (µg/mL) was determined by agar dilution assays of 361 Mongolian *H. pylori* strains. The yellow line represents the cutoff for resistance (≤4 µg/mL, sensitive; >4 µg/mL, resistant).

**Figure 3 microorganisms-11-02852-f003:**
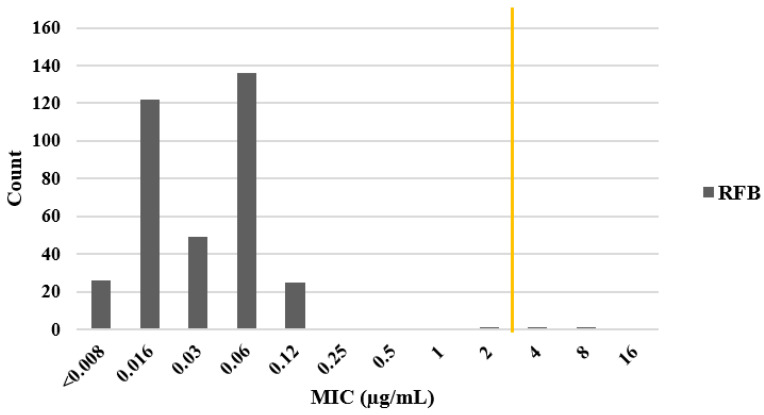
Minimal inhibitory concentration (MIC) distribution for rifabutin (RFB) against *H. pylori*. MIC (µg/mL) was determined by agar dilution assays of 361 Mongolian *H. pylori* strains. The yellow line represents the cutoff for resistance (≤2 µg/mL, sensitive; >2 µg/mL, resistant).

**Table 1 microorganisms-11-02852-t001:** Demographics and antibiotic resistance rate from a previous study and the present study.

Sex	Total Strains Tested	Antibiotic-Resistant Strains, *n* (%)						
	*n*	AMX *	CLR *	MNZ *	LVF *	MINO *	RFB	FUR
Total	361	43 (11.9)	108 (29.9)	284 (78.7)	149 (41.3)	1 (0.3)	2 (0.6)	0 (0)
Female	259	31 (11.7)	85 (32.2)	218 (82.5)	113 (42.8)	0 (0)	1 (0.4)	0 (0)
Male	94	12 (12.4)	23 (23.7)	66 (68.0)	36 (37.1)	1 (1.0)	1 (1.0)	0 (0)
Unknown gender	8	1 (12.5)	2 (25)	7 (87.5)	4 (50)	0 (0)	0 (0)	0 (0)

AMX, amoxicillin; CLR, clarithromycin; MNZ, metronidazole; LVF, levofloxacin; MINO, minocycline; RFB, rifabutin; FUR, furazolidone. * Data from our previous study [8].

**Table 2 microorganisms-11-02852-t002:** Quinolone-resistant *H. pylori* strains from 91 representative Mongolian strains.

	Total Strains Tested	Antibiotic-Resistant Strains, *n* (%)		
	*n* (%)	LVF	CIP	MOXI
Total	91 (100)	40 (44)	39 (43)	38 (42)
Female	62 (68)	28 (45)	28 (45)	27 (43)
Male	27 (30)	11 (41)	11 (41)	11 (41)
Unknown	2 (2)	1 (50)	0	0

LVF, levofloxacin; CIP, ciprofloxacin; MOXI, moxifloxacin.

**Table 3 microorganisms-11-02852-t003:** Resistance status of the two RFB-resistant strains to eight other antibiotics and their minimal inhibitory concentration (µg/mL).

No.	Strain Name	RFB	FUR	AMX	CLR	MNZ	MINO	LVF	MOXI	CIP
1	KE-9	8	0.25	0.06	0.03	32	0.06	8	4≤	4
2	UB-217	4	0.5	0.24	2	32	0.25	4	8≤	8

RFB, rifabutin; FUR, furazolidone; AMX, amoxicillin; CLR, clarithromycin; MNZ, metronidazole; MINO, minocycline; LVF, levofloxacin; MOXI, moxifloxacin; CIP, ciprofloxacin. Green box indicates resistance.

**Table 4 microorganisms-11-02852-t004:** *gyrA* gene amino acid substitutions associated with quinolone (LVF, MOXI, and CIP)-resistant *H. pylori* strains.

No.	Strain Name		MIC (µg/mL)			*gyrA* Mutation		
			LVF	MOXI	CIP	N87	A88	D91
1	Uvs	103	32	8≤	16≤	**I**		
2	Uvs	61	16	8≤	16≤	**I**		
3	UB	97	8	8≤	16≤	**I**		
4	UB	129	8	8≤	16≤	**K**		
5	UB	168	8	8≤	16≤	**K**		
6	UB	189	8	8≤	16≤	**K**		
7	Uvs	131	8	8≤	16≤	**K**		
8	Uvs	136	8	8≤	16≤	**K**		
9	Kh	160	8	8≤	16≤	**K**		
10	Go	150	8	8≤	16≤	**K**		
11	Ke	3	8	8≤	16≤	**K**		
12	Ke	47	8	8≤	16≤	**K**		
13	Ke	59	8	8≤	16≤	**K**		
14	Ke	160	4	8≤	16≤	**K**		
15	Ke	2	2	4	16≤	**K**		
16	UB	221	8	8≤	16≤		**P**	
17	UB	178	8	8≤	16≤			**N**
18	UB	198	8	8≤	16≤			**N**
19	Go	129	8	8≤	16≤			**N**
20	Uvs	97	4	8≤	16≤			**N**
21	UB	29	8	8≤	16≤			**G**
22	UB	217	4	8≤	8			**G**
23	Kh	93	2	8≤	16≤			**G**
24	UB	219	8	8≤	16≤			**Y**
25	Go	1	4	8≤	16≤			**Y**
26	Ke	1	4	8≤	8			**Y**
27	Uvs	23	4	4	8			**Y**
28	Kh	111	4	8≤	16≤			**Y**
29	Ke	87	16	1	16≤	**K**		
30	Ke	119	8	1	16≤	**K**		
31	Ke	4	0.5	8≤	16≤			**G**
32	Ke	133	8	8≤	4			
33	Uvs	113	4	8≤	16≤			
34	Kh	95	4	8≤	16≤			
35	Kh	173	2	4	8			
36	UB	30	8	0.5	0.5			
37	Ke	86	4	8≤	0.5			
38	Go	168	2	0.5	≤0.25			
39	UB	72	0.5	8≤	16≤			
Strains sensitive to quinolones (*n* = 38)						N	A	D

N, asparagine; A, alanine; Y, tyrosine; D, aspartic acid; K, lysine; P, proline; G, glycine. Orange indicates resistance to all three quinolones and green indicates resistance to one or two quinolones; blue indicates the corresponding mutations. UB, Ulaanbaatar capital city; Go, Umnugovi; Ke, Khentii; Kh, Khuvsgul; MIC, minimal inhibitory concentration; LVF, levofloxacin; MOXI, moxifloxacin; CIP, ciprofloxacin.

**Table 5 microorganisms-11-02852-t005:** *rpoB* gene amino acid substitutions associated with rifabutin-resistant *H. pylori* strains.

No.	Strain Name	MIC µg/uL	*rpoB* Mutation					
			E470	D530	V538	R701	L2196	A2710
1	Ke9	8	**G**				**P**	**V**
2	UB217	4		**N**		**C**		
3	UB75	0.12			**I**			
4	Kh67	0.06			**I**			
Sensitive stains (*n* = 217) < 0.12			E	D	V	R	L	A

E, glutamic acid; N, asparagine; D, aspartic acid; V, valine; R, arginine; L, leucine; A, alanine; G, glycine; I, isoleucine; C, cysteine. Green indicates resistance. Blue indicates the corresponding, previously reported mutation; light green indicates new hypothetical mutations. UB, Ulaanbaatar capital city; Ke, Khentii; Kh, Khuvsgul.

## Data Availability

Data are contained within the article and Appendix A.

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
