# Peer review of "Exploring Alternative Treatment Choices for Multidrug-Resistant Clinical Strains of Helicobacter pylori in Mongolia"

_microorganisms, 2023, doi:10.3390/microorganisms11122852_

Round 1

Reviewer 1 Report

Comments and Suggestions for Authors

My comments are in attachment,

Comments on the Quality of English Language

English is very good, but minor corrections may be introduced.

Author Response

Comments 1:

A total of 361 H. pylori strains isolated from patients in Mongolia were included. 

A total of 361 H. pylori strains isolated from patients in Mongolia was considered

Response 1: Thank you for We changed the word accordingly.

Comments 2:

Over the last 40 years, antibiotic resistance to Helicobacter pylori has increased globally, with higher rates in less-developed countries[1]. Because of its resistance rate, H. pylori has emerged as one of 16 antibiotic-resistant pathogens that pose the most serious threat to human health, according to the World Health Organization[2-4].

Over the last 40 years, antibiotic resistance to Helicobacter pylori was increased globally, with higher rates in less-developed countries[1]. Because of its resistance rate, H. pylori emerged as one of 16 antibiotic-resistant pathogens that pose the most serious threat to human health, according to the World Health Organization[2-4]

Response 2: Thank you for your suggestion. We changed the sentence accordingly.

Comments 3:

In Mongolia, in recent studies, it was report that multidrug resistance (MDR) in H. pylori strains is high: 51% for amoxicillin, clarithromycin, metronidazole, levofloxacin, and minocycline[14] and 60.8% for amoxicillin, clarithromycin, metronidazole, erythromycin, and nitrofuran[13]

In Mongolia, in recent studies, it was reported that multidrug resistance (MDR) in H. pylori strains is high: 51% for amoxicillin, clarithromycin, metronidazole, levofloxacin, and minocycline[14] and 60.8% for amoxicillin, clarithromycin, metronidazole, erythromycin, and nitrofuran[13].

Response 3: Thank you for your suggestion. I changed the sentence accordingly.

Comments 4:

What is the source of “unknown” people in first column, Table 1.?

Response 4: “Unknown” means “unknown gender”. Unfortunately, some participants in this study refused to check gender status in the questionary. We added “gender” in Table 1.

Comments 5:

Figure 1. Minimum inhibitory concentration (MIC) distributions for moxifloxacin (MOXI), ciprofloxacin (CIP), and levofloxacin (LVF) against H. pylori.

Figure 1. Minimal inhibitory concentration (MIC) distributions for moxifloxacin (MOXI), ciprofloxacin (CIP), and levofloxacin (LVF) against H. pylori.

Response 5: Thank you for pointing out the mistake. We corrected the word.

Comments 6:

Figure 2. Minimum inhibitory concentration (MIC) distribution for furazolidone (FUR). MIC (µg/mL) was determined by agar dilution assays of 361 Mongolian H. pylori strains.

Figure 2. Minimal inhibitory concentration (MIC) distribution for furazolidone (FUR). MIC (μg/mL) was determined by agar dilution assays of 361 Mongolian H. pylori strains

Response 6: Thank you for pointing out our mistake, and we corrected.

Comments 7:

Figure 3. Minimum inhibitory concentration (MIC) distribution for rifabutin (RFB) against H. pylori.

Figure 3. Minimal inhibitory concentration (MIC) distribution for rifabutin (RFB) against H. pylori.

Response 7: Thank you for pointing out our mistake, and we corrected.

Comments 8:

Within the gyrA gene of the 77 sequences assessed for quinolone resistance, previously reported genetic variations such as N87K/I and D91G/Y/N substitutions were frequently observed, although the A88P substitution was found in only one strain (Table 4).

Within the gyrA gene of the 77 sequences assessed for quinolone resistance, previously reported genetic variations, such as N87K/I and D91G/Y/N substitutions, were frequently observed, although the A88P substitution was found in only one strain (Table 4).

Response 8: Thank you for pointing out our mistake, and we corrected.

Comments 9:

Moreover, we found a previously described mutation (D538I) in two RFB-sensitive strains (UB75 and Kh67), with MICs of 0.12 μg/mL and 0.06 μg/mL, respectively. Moreover, we found a previously described mutation (D538I) in two RFB-sensitive strains (UB75 and Kh67), with MICs of 0.12 and 0.06 μg/mL, respectively.

Response 9: Thank you for your suggestion. I changed the sentence accordingly.

Comments 10:

For FUR, triple or quadruple regimens (100–200 mg, BID) is useable with AMX (1000 mg, BID) or tetracycline (750 mg, BID) and a PPI (standard dose, BID) with or without bismuth subcitrate (220 mg, BID) for 7–14 days[7, 10]. For FUR, triple or quadruple regimens (100–200 mg, BID) are useable with AMX (1000 mg, BID) or tetracycline (750 mg, BID) and a PPI (standard dose, BID) with or without bismuth subcitrate (220 mg, BID) for 7–14 days[7, 10].

Response 10: Thank you for your suggestion. I changed the sentence accordingly.

Comments 11:

However, it is important to note that, in accordance with previous studies, RFB and FUR-containing therapies are only recommended for individuals who have not responded to multiple (two or more) previous therapies[23-26].                   

For generality, among possible therapies, other ways could be mentioned additionally:

Response 11: Thank you for your suggestion. We removed “are only” and changed it into should be”.

3. Additional clarifications

We received confirmation of accession number from DDBJ for our genomic data. So, we modified the pre-accession numbers on page 9. We also used Tracking changes on the revised manuscript and used “yellow color highlight”.

Reviewer 2 Report

Comments and Suggestions for Authors

I have reviewed the manuscript titled “Exploring Alternative Treatment Choices for Multi-Drug Resistant Clinical Strains of Helicobacter pylori in Mongolia”. This work is well written and well designed.

The study was done on 361 H. pylori strains isolated from 1004 volunteers in Mongolia. The authors did Antibiotic susceptibility test against MOXI, 277 CIP, FUR and RFB using the serial two-fold agar dilution method to determine the minimum inhibitory concentration for 91 strains. The authors also performed DNA sequencing using the MiSeq platform to determine genetic mutation in gyrA, gyrB, and rpoB genes.

Minor comments

The date of collection of H. pylori strains was up to date. The Isolation of the H. pylori strains was between November 2014 to August 2016.

The methodology according to the reference number 14 Azzaya et al., 2020. How to isolate H. pylori strains between November 2014 to August 2016 and the methods of isolation and Antibiotic susceptibility test are in 2020.

The authors identified 361 H. pylori strains. On what criteria the authors choose 91 out of 361 H. pylori strains to did MIC. Then 91 strains were tested against FUR and RFB antibiotics. The authors did not detect any resistance in these strains so they decided to continue to test the remaining 270 strains. Please explain why you did that however testing the 270 strains did not affect the results significantly.

Author Response

Comments 1: Please add a paragraph in the discussion focused on the limitations of rifabutin treatment (such as expensive drug, myelotoxicity etc).

Response 1:

Thank you for your suggestion. Accordingly, we added the sentence on page 8, and page 4.

Comments 2: Compare your phenotype and genotype data with other studies

Response 2:

Thank you for your comments. We added the sentence on page 7.

Comments 3: add references:

             1. Pathogens 2021; 10(1):15      

             2. Microbiol Spectr 2023; 11(5): e00550

Response 3:

Thank you for your comments. We added references.

[ We added references as No.19 in the text on page 7, and as No.24 in page 8.]  

Comments 4: Check if the manuscript and the references are according to the journal guidelines

Response 4:

Thank you for pointing this out. We checked the entire references.

Reviewer 3 Report

Comments and Suggestions for Authors

In this study the authors describe the rate of resistance of 91 Helicobacter p ylori isolates from Mogolia to ciprofloxacin, moxifloxacin, furazolidone and rifabutin.  High resistance was observed to ciprofloxacin and moxifloxavin among the tested isolates. However, in vitro susceptibility of them against furazolidone and rifabutin is in concordance with previous studies, indicating that these agents could be used as alternative therapy under specific conditions. The methods and the results are well presented.

Comment:

1. Please add a paragraph in the discussion focused on the limitations of rifabutin treatment (such as expensive drug, myelotoxicity etc).

2. Compare your phenotype and genotype data with other studies.

3. add references

   Pathogens 2021; 10(1):15

   Microbiol Spectr 2023; 11(5): e00550

4. Check if the manuscript and the references are according to the journal guidelines

Author Response

Comments 1: The date of collection of H. pylori strains was up to date. The Isolation of the H. pylori strains was between November 2014 to August 2016. The methodology according to the reference number 14 Azzaya et al., 2020. How to isolate H. pylori strains between November 2014 to August 2016 and the methods of isolation and Antibiotic susceptibility test are in 2020.

Response 1:

Thank you for your comments.

Our group collected biopsy samples between November 2014 and August 2016, then the samples were transferred to Ulaanbaatar and were immediately stored at -80°C. Later, all samples were transferred for H. pylori culture and further analysis at the Department of Environmental and Preventive Medicine, Oita University Faculty of Medicine, Japan. In the first paper (Reference No. 14) from our group in 2020, we determined antibiotic resistance of 5 main antibiotics widely used in H. pylori treatment field and status of multi drug resistance. Then in this paper, we determined alternative antibiotics which are useable against multidrug resistant clinical strains.

Comments 2: The authors identified 361 H. pylori strains. On what criteria the authors choose 91 out of 361 H. pylori strains to did MIC. Then 91 strains were tested against FUR and RFB antibiotics.

Response 2:

Thank you for your comments. At first, we selected 91(25%) strains randomly from a total of 360 previous MIC determined strains. We mentioned about the selection of initial strains on page 9, and on page 3.

Comments 3: The authors did not detect any resistance in these strains, so they decided to continue to test the remaining 270 strains. Please explain why you did that however testing the 270 strains did not affect the results significantly.

Response 3:

From the representative strains, we already determined 1 resistant strain to RFB, although no resistant strain to FUR. Because Our initial purpose of this study was to find alternative rescue antibiotic against H. pylori, we need to know RFB resistant rate more precisely and need to confirm sensitivity to FUR using larger amount of strains. For this purpose, we continued to check FUR and RFB for the remaining 270 strains to see whether there is more resistance to RFB, or not to FUR. We mentioned about selection of initial strains in text on page 4.

Round 2

Reviewer 3 Report

Comments and Suggestions for Authors

In this revised version all the changes proposed have been done.